# The Synthesis of a Covalent Organic Framework from Thiophene Armed Triazine and EDOT and Its Application as Anode Material in Lithium-Ion Battery

**DOI:** 10.3390/polym13193300

**Published:** 2021-09-27

**Authors:** Shuang Chen, Shukun Wang, Xin Xue, Jinsheng Zhao, Hongmei Du

**Affiliations:** 1State Key Laboratory of Heavy Oil Processing, College of Chemical Engineering, China University of Petroleum (East China), Qingdao 266580, China; wsk971008@163.com (S.W.); xuexin@163.com (X.X.); 2College of Chemistry and Chemical Engineering, Liaocheng University, Liaocheng 252059, China

**Keywords:** lithium-ion battery, D-A type polymer, triazine, 3,4-ethylenedioxythiophene

## Abstract

As a class of redox active materials with some preferable properties, including rigid structure, insoluble characters, and large amounts of nitrogen atoms, covalent triazine frameworks (CTFs) have been frequently adopted as electrode materials in Lithium-ion batteries (LIBs). Herein, a triazine-based covalent organic framework employing 3,4-ethylenedioxythiophene (EDOT) as the bridging unit is synthesized by the presence of carbon powder through Stille coupling reaction. The carbon powder was added in an in-situ manner to overcome the low intrinsic conductivity of the polymer, which led to the formation of the polymer@C composite (PTT-O@C, PTT-O is a type of CTFs). The composite material is then employed in LIBs as anode material. The designed polymer shows a narrow band gap of 1.84 eV, proving the effectiveness of the nitrogen-enriched triazine unit in reducing the band gap of the resultant polymers. The CV results showed that the redox potential of the composite (vs. Li/Li^+^) is around 1.0 V, which makes it suitable to be used as the anode material in lithium-ion batteries. The composite material could exhibit the stable specific capacity of 645 mAh/g at 100 mA/g and 435 mAh/g at 500 mA/g, respectively, much higher than the pure carbon materials, indicating the good reversibility of the material. This work provides some additional information on electrochemical performance of the triazine and EDOT based CTFs, which is helpful for developing a deep understanding of the structure–performance correlations of the CTFs as anode materials.

## 1. Introduction

Although commercial lithium-ion battery (LIB) electrodes are mainly inorganic materials, these materials still have their application limitations, such as low specific capacity, low elemental abundance, heavy metal issues and so on [1,2,3]. The structure of inorganic materials is relatively fixed, so that it is difficult to increase their energy density through some flexible methods [4]. Compared with inorganic materials, organic materials have many advantages, including versatile and flexible structural tunnability through molecular engineering to achieve some specific functions [5,6] and the widely available raw materials. In addition, in order to maintain the specific crystal structure and the high purity of inorganic materials, harsh reaction conditions such as high annealing temperature are often required, while the conditions for the synthesis of organic materials are relatively mild [7]. Therefore, in the recent years, research on organic electrodes has regained people’s interest.

Compared with organic small molecular electrode materials, conjugated polymers have attracted more extensive attentions due to their non-dissolution property [8,9], higher conjugation length, lower band gaps, higher conductivity, etc. [10]. A main limitation for organic electrode materials is their lower conductivity, which makes their active sites unavailable to lithium ions as well as electrolytes [11,12]. The conjugated compounds containing N atoms can be used as electrochemical active materials in alkali metal rechargeable batteries, and the introduction of the N atoms can help the materials to reduce the band gaps, increase electrical conductivity, and improve the redox activity of the materials [13]. Chen et al. prepared a π conjugated COF based on the reaction of 2,4,6-triaminopyrimidne and 1,4-phthalaldehyde, which showed a high capacity of up to 401.3 mAh g^−1^ at 1 A g^−1^ and also showed excellent long-term cyclability [14] as an anode electrode. Kim et al. synthesized several network polymers with two-dimensional nanosheet morphology by Stille coupling reaction under solvothermal condition, among which, the CON-16 material derived from 2,4,6-tris(thiophene-2-yl)-1,3,5-triazine (as the building block) and thieno[3,2-b]thiophene (as the bridge unit) possessed a capacity of ~250 mAh g^−1^ at 100 mA/g, used as the anode material for sodium-ion batteries [15]. The long-conjugated structures in the polymers will promote the distribution of the π electron cloud throughout the whole molecular skeleton, facilitating the electron delocalization and effectively reducing the band gap (*E*_g_) of organic materials [16]. The three N atoms in the triazine unit endowed the thiophene armed triazine monomer with the electron deficient character and the thiophene derivatives, as the bridging unit has the characteristic electron donating property. As a general rule, the alternating combination of an electron deficient unit and an electron donating unit by specific polymerization method will lead to reduced band gap, elevating the conductivity of the resultant polymer [17]. These characteristics make the triazine based conjugated polymers be particularly good as anode materials because the highly required low redox potentials are just derived from the low band gaps of the polymers [18,19]. The composite materials formed between conjugated polymers and carbon materials, including carbon nanotube, graphene or carbon powder, can overcome the low conductivity of the polymers and make active the sites of the polymer materials to the lithium ions and electrolytes as well [20]. Wang et al. prepared a composite material (COF@CNTs) by the in-situ polymerization of 1,3,5-benzenetricarboxaldehyde and p-phenylenediamine on the surface of carbon nanotube (CNT), which had a capacity of 1536 mAh g^−1^ [12]. For a single repeating unit of the COF, the lithium storage sites include both C=N and C=C bonds [12].

More recently, we synthesized series of conjugated organic framework (COF) materials based on triazine and thiophene derivatives [21]. The triazine has a strong electron deficient property, while thiophene and its derivatives have strong electron donating properties, which endowed the COFs with donor-acceptor type characters and low band gaps, which makes them be suitable as anode materials for lithium batteries. It was interesting to find that the conjugation length of the bridge units has an effect on the lithium storage capacity of the resultant polymers. The composite PTT-4@C derived from the in-situ polymerization of thiophene armed triazine (bromo-compounds) and the organotin terminated thieno[2′,3′:4,5]thieno[3,2-b]thieno[2,3-d]thiophene (TTTT) by the presence of carbon powder delivered the highest specific capacity of 772 mAh g^−1^. However, in this work, an important thiophene derivative, 3,4-ethylenedioxythiophene (EDOT), which is widely used as an electron donor unit to construct D-A type conjugated polymers, was ignored.

Herein, we adopted the thiophene armed triazine (2,4,6-tris (5-bromothiophen-2-yl)-1,3,5-triazine) (TBYT) as the as the building unit, and the organotin terminated EDOT as the bridging unit for the construction of the COFs (PTT-O), as shown in Figure 1. The polymerization was conducted by the presence of carbon powder in an in-situ manner with the aim to enhance the conductivity of the composite, and the resulting composite PTT-O@C was assessed in terms of its lithium storage capacity, long term operation stability and redox activity.

## 2. Experimental

### 2.1. Synthesis of the Monomer TBYT

The reagents and their purchasing channels can be referred to in the Appendix A. In total, 7.02 g of trifluoromethanesulfonic acid (TfOH, Aladdin Co., LTD, Shanghai, China) and 9.03 g of 5-bromothiophene-2-carbonitrile (BTCN, Zhengzhou Alfa Chemical Co., Ltd, Zhengzhou, China) were added to 120 mL of chloroform in a flask at 0 °C with constant stirring for 5 h. Then, the mixture solution was further stirred for another 48 h at 30 °C. After that, chloroform was distilled, and the residue was neutralized with ammonia. The crude solid product was washed with excessive water and ethanol and then dried overnight. The raw TBYT was further purified by the recrystallization method in toluene twice, and the final product was dried in vacuum oven for ten hours and obtained as off-white color powder (TBYT) [21].

### 2.2. The Synthesis of PTT-O and PTT-O@C

The synthetic procedure for PTT-O can be referred to in our latest work [21]. For this, 300 mg of TBYT, 373.2 mg of 2SnEDOT (SunaTech Inc, Suzhou, China), 28 mg of PdCl_2_(PPh_3_)_2_ (Aladdin Co., LTD, Shanghai, China) and 80 mL of toluene were used as the reaction mixture. The PTT-O@C was synthesized by adding 667 mg of Vulcan XC-27 carbon powder (C) (Aladdin Co., LTD, Shanghai, China) to the above mixture. The reaction condition and purification procedure are identical to our latest work [21]. The weight ratio of the polymer accounts for 30% of the PTT-O@C composite.

### 2.3. The Detailed Information on Materials and Characterization

The materials used, the detailed information on characterization techniques and the electrochemical measurements are given in the Appendix A. A schematic diagram of the coin-type battery is given in Appendix A.

## 3. Results and Discussion

### 3.1. Materials Characterization

Figure 2 shows the SEM images (Hitachi Su-70, Hitachi Inc., Tokyo, Japan) of PTT-O (Figure 2a) and PTT-O@C (Figure 2b). The particles of PTT-O (Figure 2a) were in a slender strip-like stacked state, and the whole material grew irregularly. When the polymer was composited with the carbon powder, it was difficult to clearly distinguish the polymer phase and the carbon powder particle (Figure 2b). The composite showed a loose porous structure formed by the agglomeration of small particles, and the uniform elemental mapping distribution of C, N and S indicated that PTT-O had been deposited on the carbon particles (Figure 3). It is assumed that the presence of carbon particle can help the dispersion of the newly formed polymer, making the polymers grow gradually on carbon powder [22].

Figure 4a shows the FT-IR spectrum (Nicolet Avatar 360 FT-IR, Thermo Fisher Scientific, Waltham, MA, USA) of PTT-O polymer. The peak at around 3450 cm^−1^ is attributed to the vibration of O-H derived from the trace water in the polymers. The peak at 790 cm^−1^ is the bending vibration (out of plane) of C_β_-H, and the peak at 1038 cm^−^^1^ can be assigned to the bending vibration (in plane) of C_β_-H in thiophene groups. Moreover, another peak at 1039 cm^−1^ is the stretching vibration (SV) of C-O in EDOT unit. The peak observed at around 1367 might be due to the SV of C=N in the triazine ring, and the peaks at 1426, 1490 and 1653 cm^−1^ are attributed to the skeletal vibration of the thiophene or EDOT rings [21]. Figure 4b shows the thermogravimetric (TG, Netzsch STA449C TG/DSC, NETZSCH Scientific Instruments Trading Ltd., Waldkraiburg, Germany) curves of PTT-O. The sample was heated from 20 °C to 650 °C under nitrogen atmosphere, and the heating rate was 5 °C/min. From the room temperature to 400 °C, the TG curve shows a stable thermal platform until the decomposition temperature. The weight loss was very minor until the temperature reached to 400 °C. The decomposition temperature (T_d_) of PTT-O is 529.6 °C, at this point the weight loss was 5%, indicating the robust stability of the polymer, which fully meets the application requirements for electrode materials [16].

An XPS survey (ESCALAB 250Xi, Thermo Fisher Scientific, Waltham, MA, USA) was conducted to study the elemental compositions and the valence states of PTT-O@C. Figure 5 shows the XPS spectroscopy of C, N and S elements for the composite. The C 1s peaks at 284.6, 285.7 and 286.8 eV belong to the sp^2^ C=C in the thiophene (or EDOT) rings, the sp^2^ C-N in the triazine group and the sp^2^ C-S in the thiophene rings, respectively. The splitting peaks including S 2p_1/2_ at 164.24 eV and S 2p_3/2_ at around 163.12 eV are the sp^2^ S in thiophene rings, and the N 1s peak at 397.8 eV is due to the pyridinic-N from the triazine unit. The peak of O 1s at 532.5 eV is the O-C in EDOT unit [21].

Figure 6a shows the ultraviolet-visible diffuse reflectance spectra (Varian Carry 5000, Agilent Technologies Ltd, Mulgrave, Australia) of PTT-O. The band gap (*E*_g_) of indirect semiconductors can be calculated by the following formula [23]:(αhν)^1/2^ = K (hν − *E*_g_)

Among them, α is the absorption coefficient, h is the Planck constant, ν is the vibration frequency, and K is the general constant. If (αhν)^1/2^ is used to plot the graph against the hν, and the straight-line part in the figure is extended to the X axis, then the intercept of the curve on the X axis is the band gap *E*_g_ of the indirect semiconductor. Figure 6b shows the Tauc plot of PTT-O, and the band gap of PTT-O is calculated to be 1.84 eV. This proves that the combination of the triazine unit and the EDOT unit in a D-A type manner can effectively reduce the band gap of organic semiconductors. Moreover, compared with the thiophene unit, the EDOT unit has an ethylenedioxy unit on the 3,4 position of the thiophene ring. Among the O atoms, which have a sp^3^ hybridization character, two orbitals participate in the formation of two σ bands and each oxygen atom has two pairs of lone pair electrons, leaving two pairs of lone pairs of electrons participating in the conjugate system, endowing the EDOT unit with stronger electron donating ability than that of thiophene. As a result, the copolymer resulting from the copolymerization of the EDOT unit and the triazine containing unit (TBYT) have a lower band gap than that of the corresponding copolymer employing thiophene as the bridging unit. In this case, the copolymer PTT-O might have a higher conductivity than that of the PTT-1 reported in our latest work [21].

The pore structure of PTT-O@C was measured by nitrogen isotherm adsorption-desorption at 77.3 K. The adsorption-desorption curves are shown in Figure 7. It can be seen that the composite material exhibits a combination curve with type IV isotherms and H3-type hysteresis loop. The sharp increase of the curve under high relative pressure is because of the capillary condensation in the slit pores of the composite formed by the disordered packing between composite particles, indicating that the composite is a mesoporous material (Figure 7a). The pore size distribution in Figure 7b further illustrates that there are mainly mesopores in the PTT-O@C composite. The specific surface areas of Vulcan XC-72 carbon and PTT-O@C are 173.0722 m^2^/g and 71.0110 m^2^/g, respectively. Apparently, the coating of the PTT-O polymer on the carbon powder lead to the decrease of the specific area of the carbon powder, which might due to the coverage of a portion of its original pore structures of the carbon powder UV-Vis absorption [24].

### 3.2. Electrochemical Properties

Figure 8 shows the CV curves (PGSTAT 302N, Metrohm, Herisau, Switzerland) of PTT-O@C in a coin type lithium-ion battery. For this, the scan rate was 0.1 mV/s, starting from open-circuit voltage of the battery. The counter electrode in the coin-type cell was lithium slice, and the voltage window was 0–3 V. When scanning towards the cathodic direction, the reduction reactions firstly occurred on the working electrode. It can be seen that during the first cathodic scan, two reduction peaks appeared at 0.67 V and 0 V, which corresponded to the formation process of SEI film (solid electrolyte film) on the surface of the electrode material [25]. In the subsequent scanning process, these two reduction peaks disappeared, indicating that the SEI film had been formed on the surface of the electrode material after the first cycle, existed stably during the subsequent charge and discharge processes and did not participate in the subsequent electrochemical reaction (Figure 8a). In the process of scanning towards the anodic direction, the material underwent an n-type undoping reaction, that is, lithium ions were extracted from the composite material and entered into the electrolyte. The intercalation of lithium ions in the carbon-based materials corresponds to the broad peaks at around 0 V, and the pair of redox peaks between 0.7 V and 1.2 V correspond to the lithium-ion deintercalation process in the polymer matrix accompanying an n-type undoping reaction [21]. In addition, another oxidation peak, which appeared at about 2.5 V, was also observed on the curves; this is the p-type doping process on the polymer, that is, when all the lithium ions in the polymer had been deintercalated, the PF6^-^ in electrolyte entered into the polymer structures, forming the so-called p-type doping [21]. Figure 8b shows the CV curves of PTT-O@C with different scan rates; the position of peaks at around 1 V did not change with the change of scan rate, illustrating the good reversibility of the battery.

Figure 9a shows the voltage-specific capacity curve during constant current charge and discharge of PTT-O@C. The attenuation of specific capacity during the first and second discharge originates from the formation of the SEI film. This process can also be seen from the two discharge platforms on the first cycle discharge curve, which could correspond to the two reduction peaks on CV curve. On the charging curve, another charging stage at about 2.5 V can also be observed, which is the p-type doping process given by the anodic scan on the CV curve at about 2.5 V. After the first discharge process, the charge-discharge curve of the material no longer showed any obvious charge-discharge platform, meaning that during the charging/discharging process the material only changed its charge state.

Figure 9b,c is the cyclic discharge specific capacity curve and the Coulombic efficiency curve of PTT-O@C with the current densities of 100 mA/g and 500 mA/g respectively. It can be seen that the material has a large specific capacity loss between the first and the second discharge, which is caused by the formation of the SEI film during the first discharge process [19]. At the same time, the formation of SEI film also caused the low coulombic efficiency of the material on the first cycle, which was 45.9%. After that, the coulombic efficiency of the material was stabilized and approached 100%, indicating the material’s good reversibility during the charge-discharge cycling process. In addition, the specific capacity of the composite increases with the increase of the running cycles, which is especially obvious at large currents. The reason is that many active sites are deeply embedded within the electrode material, and with the continuous intercalation and deintercalation of lithium ions, the structure of the polymer is gradually opened so that more and more hidden active sites are slowly exposed to lithium ions and electrolyte due to the formation of a new ion transport channel. And, as a result, the specific capacities showed a gradual upward trend at the initial 300 cycles [26]. At 100 mA/g, PTT-O@C exhibited the specific capacity of 645 mAh/g after 300 cycles, which is much higher than the specific capacity of Vulcan XC-72 carbon (the black line in Figure 9c). Besides, no decrease in specific capacity was observed after 300 cycles, which proves that the material has good performance at low current densities. When the current density reached 500 mA/g, the specific capacity of the material rose sharply at the first 300 cycles and then became stable with the specific capacity of around 435 mAh/g. Figure 9d shows the rate performance of PTT-O@C at different current densities, and it can be seen that PTT-O@C could recover a high specific capacity level during the recovery from a large current density of 5000 mA/g to a small current of 100 mA/g, which reflects the good rate performance of the PTT-O@C. The performance of PTT-O@C is compared with other CTFs@C composites possessing similar structures. As shown in Appendix A, the PTT-O@C has a higher capacity than that of the PTT-1@C. However, the specific capacity of PTT-O@C is lower than the other three composites, PTT-2@C, PTT-3@C and PTT-4@C. The linkage of the 3,4-ethylenedioxy- unit on the thiophene ring enhanced the electron donating ability of the EDOT unit and then extended the conjugation length of the resultant polymer of PTT-O, and as a result, the PTT-O@C composite has a higher capacity than that of the PTT-1@C composite, which is consistent with the results reported previously [21].

Figure 10 shows the Nyquist curve of the AC impedance test of PTT-O@C. The frequency range for the test is from 100,000 Hz to 0.01 Hz. Figure 10a is the impedance spectrum measured under the open circuit voltage, which consists of a small arc and an incomplete large arc, in the high frequency region and in the middle and low frequency region, respectively [27]. The small reactance arc indicates the transport process of Li^+^ in the passivation layer, which is on the top of the electrode material, and the incomplete large arc corresponds to the electric double layer capacitor during the charging and discharging process. At the open circuit voltage, the deintercalation process for lithium ion has not yet occurred, the charge transfer resistance is relatively large, so a capacitive reactance arc with a large radius is shown on the Nyquist curve (the big arc in Figure 10a). The resistance of the electric double layer is also called the charge transfer resistance (R_ct_), and it is an important parameter to characterize the transfer kinetics of the lithium ions, which can be obtained from the intercept of the big arc on the X axis, as shown in Figure 10a. In this case, the R_ct_ value for the PTT-O@C electrode at open-circuit voltage is about 142 Ω. Figure 10b is the AC impedance spectrum measured under the polarization voltage. The inset in the image is the equivalent circuit diagram after fitting the spectrum [28]. It can be seen that the Nyquist curve of the electrode at the biased potential consists of a small arc, a large arc and straight line in turn, in the high, middle and low frequency regions, respectively. Among them, the first small arc indicates the transport process of lithium ions in the SEI film on the electrode surface. The large capacitive reactance arc in the middle frequency region indicates the transportation of lithium ions at the interface of the electric double layer formed between the SEI film and the electrode material. The straight line, located in the last part and called the Warburg resistance, indicates the solid phase transport of lithium ions. Based on the above analysis, the equivalent circuit corresponding to the curve can be established, as shown in the inset in Figure 10b [28] where R_Ω_ is equivalent to the ohmic resistance of the system, including the ohmic resistance of the electrolyte, the diaphragm and the electrode material itself. Moreover, Q_SEI_ and R_SEI_ are two constant phase components, indicating the capacitance and resistance of the SEI film, respectively. The Q_d_ represents the capacitance of the electric double layer, and R_ct_ is the resistance of the electric double layer, which is also called the charge transfer resistance. The Q_w_ indicates the solid phase transport of lithium ions, which is called the solid-phase diffusion impedance. Considering that the R_Ω_ in the equivalent circuit is greatly affected by the electrolyte diaphragm and other components in the circuit, and the electrode process mostly occurs on top of the electrode material, the charge transfer resistance, R_ct_, of the electrode material is usually used to characterize the charge transfer ability of the material. According to the fitting data, the R_ct_ value of the electrode is 56.4 Ω for the PTT-O@C electrode at polarization potential, proving the good electrical conductivity of the material [29].

## 4. Conclusions

In this study, we adopted TBYT as the building block and EDOT as the bridging unit to synthesize a triazine-based COF named as PTT-O through Stille coupling reaction. Based on this, a composite materials PTT-O@C was obtained by the in-situ polymerization of the polymer on the carbon powder, and the composite material was then used as the active material of the anode electrode in a lithium-ion battery. The SEM results show a homogeneous composite structure is formed for the PTT-O@C, which is composed of small particles agglomerated together to form a porous structure that is conducive to the migration of lithium ion and electrolyte. The polymer PTT-O bearing EDOT as the bridging unit has a lower band gap (1.84 eV) than that of the homologues COF (PTT-1) bearing the thiophene unit (1.92 eV) due to the higher electron donating ability of EDOT compared to thiophene. From the CV of the electrode material in the half-cell, the potential for the n-type doping/undoping process of lithium ions in PTT-O@C is about 1 V (vs. Li/Li^+^), and the formation of the SEI (Solid Electrolyte interphase) film can be observed during the first discharge of the material. As the battery was fully activated for the charging/discharging cycles, the specific capacities of PTT-O@C are 645 mAh/g and 435 mAh/g at the current density of 100 mA/g is and 500 mA/g, respectively.

## Figures and Tables

**Figure 1 polymers-13-03300-f001:**
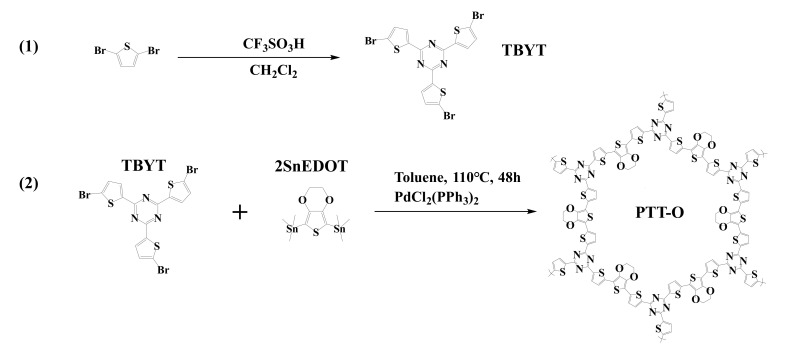
The synthesis route of TBYT (**1**) and PTT-O (**2**).

**Figure 2 polymers-13-03300-f002:**
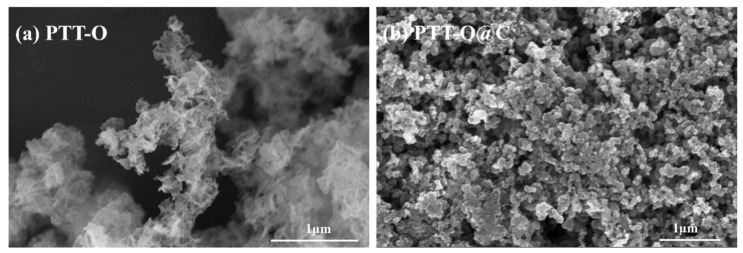
The SEM images of (**a**) PTT-O and (**b**) PTT-O@C.

**Figure 3 polymers-13-03300-f003:**
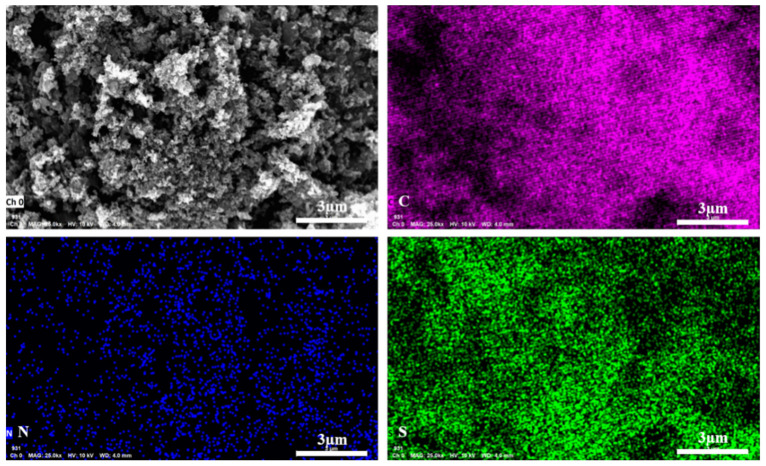
The elemental mapping of C, N and S of PTT-O@C.

**Figure 4 polymers-13-03300-f004:**
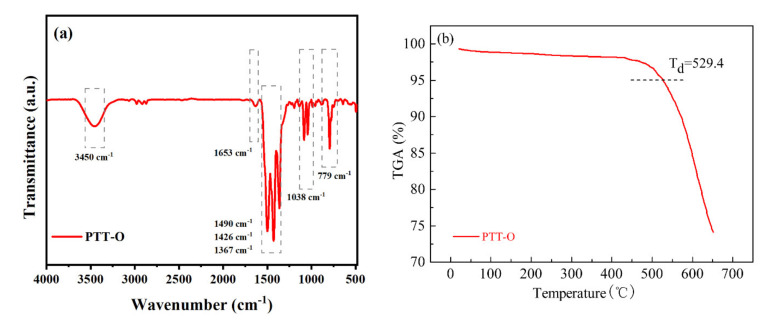
(**a**) The FT-IR spectrum of PTT-O; (**b**) the Thermogravimetric curves of PTT-O.

**Figure 5 polymers-13-03300-f005:**
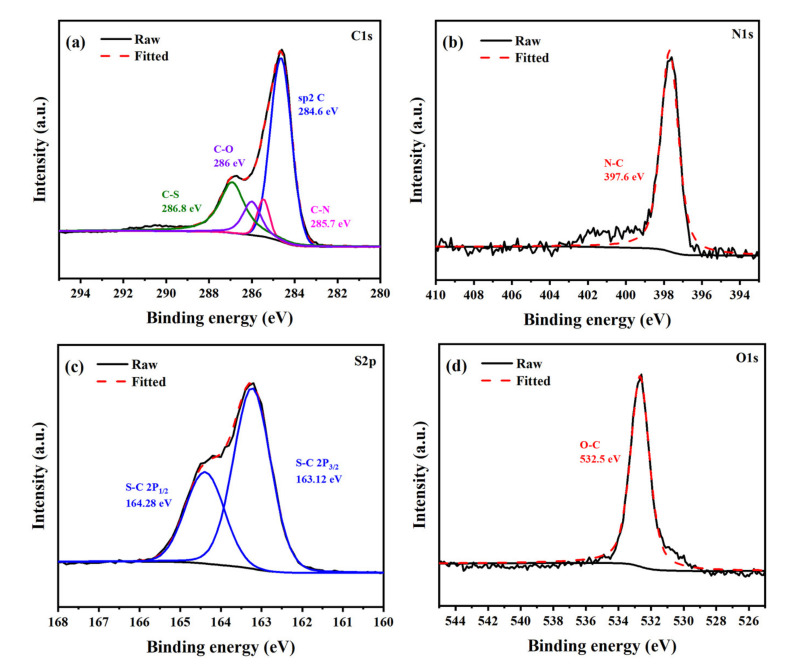
The High-resolution XPS spectra of PTT-O@C (**a**) C 1s, (**b**) N 1s, (**c**) S 2p and (**d**) O 1s.

**Figure 6 polymers-13-03300-f006:**
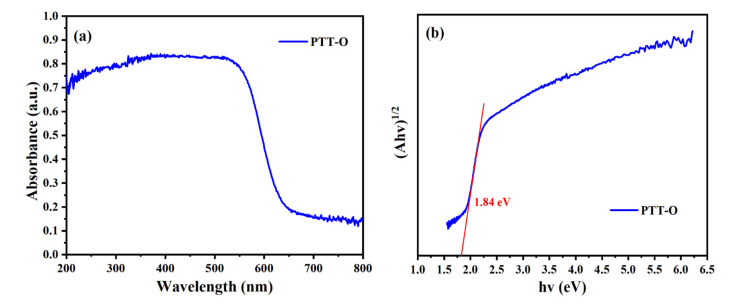
(**a**) The UV-VIS spectrum of PTT-O; (**b**) the Tauc plot of PTT-O.

**Figure 7 polymers-13-03300-f007:**
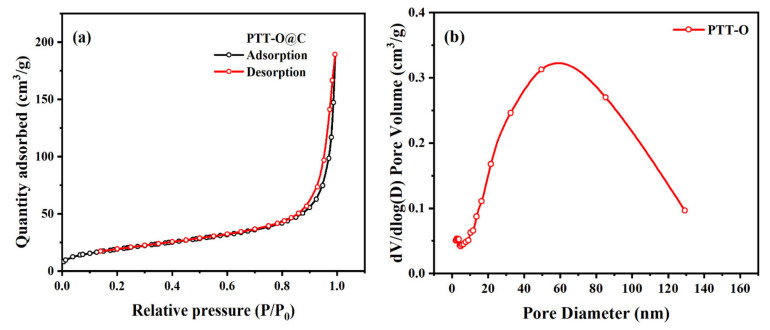
(**a**) The N_2_ adsorption and desorption curve of PTT-O@C; (**b**) the pore size distribution of PTT-O@C.

**Figure 8 polymers-13-03300-f008:**
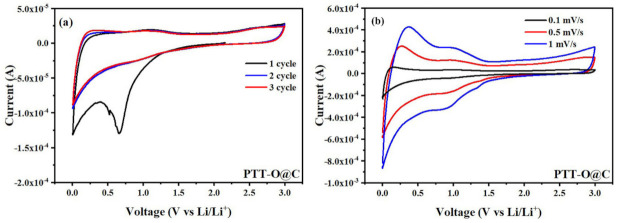
(**a**) CV curve of PTT-O@C at 0.1 mV/s; (**b**) CV curve of PTT-O@C at different scan rate.

**Figure 9 polymers-13-03300-f009:**
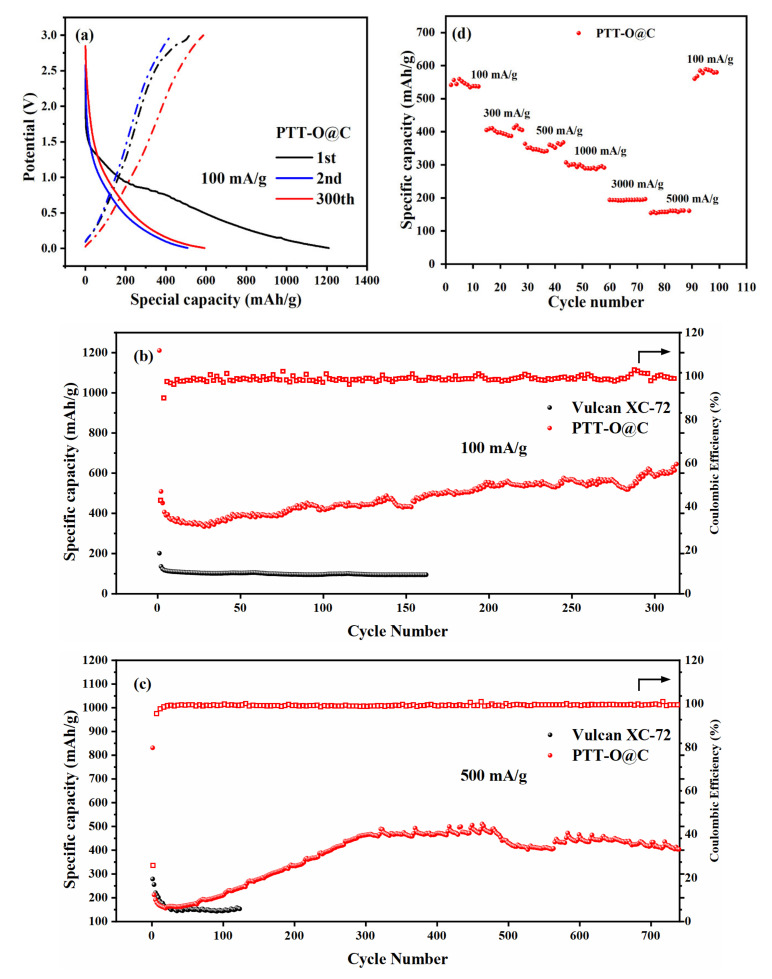
(**a**) The GDC curve of PTT-O@C; (**b**) the cycling performances of PTT-O@C at 100 mA g^−1^, the solid red ball represents the specific capacity of PTT-O@C, and the black ball represents the specific capacity of Vulcan XC-72, the hollow box with red color represents coulombic efficiency (%) of the PTT-O@C; (**c**) the cycling performances of PTT-O@C at 500 mA g^−1^, the solid red ball represents the specific capacity of PTT-O@C, and the black ball represents the specific capacity of Vulcan XC-72, the hollow box with red color represents coulombic efficiency (%) of the PTT-O@C; (**d**) the rate performance of PTT-O@C.

**Figure 10 polymers-13-03300-f010:**
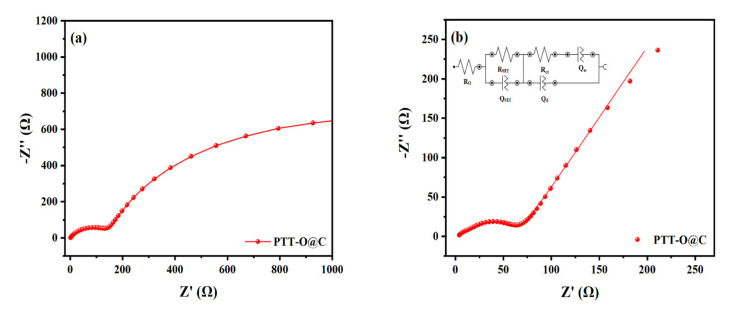
EIS spectra of PTT-O@C (**a**) at open-circuit voltage (**b**) at polarization potential.

## Data Availability

All the data is available within the manuscript.

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
