# Peer review of "The Synthesis of a Covalent Organic Framework from Thiophene Armed Triazine and EDOT and Its Application as Anode Material in Lithium-Ion Battery"

_polymers, 2021, doi:10.3390/polym13193300_

Round 1
Reviewer 1 Report
This work reports the preparation of triazine/thiophene-based covalent organic framework (COF) with ethylenedioxythiophene (EDOT) bridging unit. The COF/carbon composite delivered specific capacity of 645 mAh/g at 100 mA/g and 435 mAh/g at 500 mA/g. This is a technically valid report. Specific comments are given below.
- The title is unnecessarily lengthy. It is not recommended to use sentence-like style with comma. The current abstract is more like an experiment summary, and it should be re-written. The background and significance of this work should be clearly demonstrated.
- Is there Fig. 1 in the manuscript?
- This study is a close follow up of the authors’ previous work (ref. 21). The novelty and significance of the present study is not demonstrated as the structural design of COF is so similar.
- The authors claim that FTIR and TGA data indicate residue water and solvent. These could be strong interfering factors for anode application. It should be demonstrated that these species are removed in the final electrode sample.
- The authors claim that the specific capacity increase in long-term cycling because the structure of the polymer is gradually opened so that more and more hidden active sites were slowly exposed to lithium ions and electrolyte. Does this indicate the material synthesis/electrode fabrication is far from optimized? How does it correspond to the nitrogen isotherm adsorption-desorption data?
- This work is a very close follow-up study of the authors’ recent report of very similar COF system (ref 21). The performance of the new materials in this study should be compared to the previous study. Based on the comparisons, the benefit of the new structure design should be explained and validated with experimental data.
Reviewer 2 Report
The work titled "The synthesis of a covalent organic framework from thiophene armed triazine and EDOT, and its composite with carbon used as anode material in lithium-ion battery" is an interesting work with lot of potentials for future. I believe this article should be accepted after the following changes-
- Figure number needs to be corrected. The article started with figure 2.
- What is the origin of thermal degradation of EDOT at such high temperature?
- The abbreviations like PTT or PTT-O/C etc. needs to be clarified once and then they can be used.
- Figure 10- b and c: the red solid and hollow dots needs to be clarified in the text or in the image. It is quite confusing now. Authors should use proper legends.
- What is the charge transfer resistance value?
- Authors should provide a physical image of their actual cell they have used.
- There should be a table of comparison with recent research vs the data obtained here.
- Authors may also conduct a post 300 cycle XPS to understand the surface behavior of the composite to provide an argument against. such high capacitance retention
Reviewer 3 Report
This paper presents a very interesting work on new materials for batteries.
I have only one question regarding the efficiency of the new polymers under temperature variations in the -20 ...+ 50 C range.
Round 2
Reviewer 2 Report
The article can be accepted.